# Longitudinal Outcomes Following Mitral Valve Repair for Infective Endocarditis

**DOI:** 10.3390/microorganisms12091809

**Published:** 2024-09-01

**Authors:** Yuan Qiu, Lawrence Lau, Zaim Khan, David Messika-Zeitoun, Marc Ruel, Vincent Chan

**Affiliations:** 1Division of Cardiac Surgery, University of Ottawa Heart Institute, Ottawa, ON K1Y 4W7, Canada; yqiu@ottawaheart.ca (Y.Q.); zkhan@ottawaheart.ca (Z.K.);; 2Division of Cardiology, University of Ottawa Heart Institute, Ottawa, ON K1Y 4W7, Canada

**Keywords:** mitral valve, infective endocarditis, mitral valve repair, longitudinal outcomes, mitral surgery

## Abstract

Mitral valve repair is the ideal approach in managing mitral valve infective endocarditis for patients requiring surgery. However, viable repair is influenced by the extent of valve destruction and there can be technical challenges in reconstruction following debridement. Overall, data describing long-term outcomes following mitral repair of infective endocarditis are scarce. We, therefore, assessed the late outcomes of 101 consecutive patients who underwent mitral valve repair for IE at the University of Ottawa Heart Institute from 2001 to 2021. The 5- and 10-year survival rate was 80.8 ± 4.7% and 61.2 ± 9.2%, respectively. Among these 101 patients, 7 ultimately required mitral valve reoperation at a median of 5 years after their initial operation. These patients were of a mean age of 35.9 ± 7.3 years (range 22–44 years) at the time of their initial operation. The 5- and 10-year freedom from mitral valve reoperation was 93.6 ± 3.4% and 87.7 ± 5.2%, respectively. Overall, mitral valve repair can be an effective method for treating infective endocarditis with a favourable freedom from reoperation and mortality over the long term.

## 1. Introduction

Infective endocarditis (IE) is associated with morbidity and mortality ranging between 10 and 30% [1,2]. There is a rising incidence of IE in North America owing to an aging population and an increase in risk factors such as intravenous drug use [3,4,5]. In 25–30% of cases, surgery is indicated in addition to antimicrobial therapy, although surgery is associated with its own risks and complications [2]. For patients with IE involving the mitral valve, valve repair, when possible, is favoured owing to issues related to prosthetic heart valves, risk of future re-infection of prosthetic material, and the need for anticoagulation [6,7,8,9]. However, it is technically challenging and dependent on the amount of native tissue available for reconstruction once infected tissue is resected and debrided [8,10]. Mitral valve repair should be carried out depending on the extent of destruction from IE, acuity of the disease, and individual patient characteristics [11,12]. The 2023 European Society of Cardiology (ESC) guidelines for the management of endocarditis state that mitral valve repair cannot be concluded as superior to replacement due to the high probability of selection bias in the literature [6,12,13,14]. They state that preservation of the valve in acute IE should only be attempted if there can be full eradication of infected tissue with a durable repair achieved [12]. Currently, the research in the literature reporting outcomes of mitral valve repair for endocarditis includes small cohorts of patients with limited long-term follow-up [15]. Our study aims to investigate the longitudinal outcomes, such as survival and durability, following mitral valve repair for native valve endocarditis at our institution.

## 2. Materials and Methods

### 2.1. Study Design and Patient Selection

This was a retrospective, single-center study including patients who underwent mitral valve repair surgery for mitral valve IE between 2001 and 2021 at the University of Ottawa Heart Institute. Overall, 101 consecutive patients undergoing complex repair at our institution were included in this analysis. All operations were performed by 2 surgeons with extensive expertise and training in mitral valve repair. Preoperative, intraoperative, and postoperative data were retrieved from a prospectively collected database as part of the Surgical Mitral Valve Database, with clinical follow-up previously described, including echocardiographic follow-up [16]. Ethics approval was obtained through the University of Ottawa Heart Institute Research Ethics Board (approval number 20160395-01H) and is active for the longitudinal assessment of outcomes following mitral valve repair. Informed consent was obtained for all patients undergoing surgery. Follow-up averaged 4.4 ± 3.8 years, ranged up to 15.8 years, and was completed for all patients.

### 2.2. Statistical Analysis

Continuous variables are expressed as the mean ± standard deviation or as the median with an interquartile range if skewed. Categorical variables are described as numbers (percentages). Survival and freedom from reoperation were assessed using the Kaplan–Meier method and predictors of risk were assessed with the separate semi-parametric Cox proportional hazards model, with *p* < 0.05 considered significant.

## 3. Results

### 3.1. Preoperative Demographics

A total of 101 patients were included. The mean age was 55.3 ± 14.2 years, and 33 (33%) of the patients were female. Of these, 13 (13%) had atrial fibrillation, and 13 (13%) patients were in New York Heart Association (NYHA) class III or IV heart failure symptoms preoperatively. The mean risk of mortality as estimated by the Society of Thoracic Surgeons risk calculator was 1.6%. There were 8 (8%) patients with mitral perforation, and 83 (82%) patients had grade III or IV mitral regurgitation preoperatively. Nine (9%) patients required concomitant coronary artery bypass grafting. The preoperative patient characteristics are summarized in Table 1.

### 3.2. Causative Microorganisms 

The causative microorganism was *Streptococcus viridans* in 21 (21%) of cases, and *Staphylococcus aureus* in 7 (7.0%) of cases. Twenty-two (22%) patients had negative blood cultures due to previously healed IE (in twenty-one cases) and one patient had Libman Sacks endocarditis. The remaining blood culture microorganisms are summarized in Table 2. One patient with Coagulase-negative *Staphylococci*-positive blood cultures had a previous bioprosthetic aortic valve. Patients were treated with the appropriate antimicrobial therapies perioperatively. 

### 3.3. Surgical Indication and Technique

Infective endocarditis was diagnosed based on the Duke criteria considering assessment by the treatment team [17]. Of all the patients who underwent mitral valve repair, 89 (88%) patients underwent surgery due to heart failure with mitral regurgitation. Two (2.0%) had persistent infection (one with large vegetation despite antibiotic management and one with persistent fever despite antibiotics). Six patients (5.9%) had locally uncontrolled infection; four of these patients had aortic root abscess, one had a fistula through the mitral valve annulus from the left ventricle to the left atrium, and one had a left ventricular false pseudoaneurysm below the aortic root. Half of the four patients who had septic emboli had cerebrovascular accidents secondary to the septic emboli. Main indication for surgery is summarized in Table 3. Indications for surgery were discussed with an interdisciplinary IE team and aligned with the 2023 European Society of Cardiology guidelines for the management of endocarditis [12].

Patients predominantly underwent mitral valve repair with ring annuloplasty, although six patients did not receive an annuloplasty band. The most used annuloplasty was the Futureband, which was implanted in 95 (Medtronic, Minneapolis, MN, USA), whereas 4 patients underwent annuloplasty with the Physio ring (Edwards Lifesciences, Irvine, CA, USA). The median annuloplasty size was 30 mm. In addition to this, patients predominantly received leaflet resection involving either the anterior (n = 13) or posterior (n = 25) leaflet. Twenty-five patients underwent chordal replacement following debridement and twenty-one underwent in situ chordal transfer. The surgical techniques used for mitral valve repair are summarized in Table 4. 

### 3.4. Short- and Long-Term Survival

Perioperative mortality occurred in one patient (1%) within 30 days of surgery. Overall, five-year survival was 80.8 ± 4.7% and ten-year survival was 61.2 ± 9.2%. The Kaplan–Meier survival curve is displayed in Figure 1. Risk factors associated with mortality are shown in Appendix A. While there was a trend towards worse survival amongst females and in those with preoperative atrial fibrillation, there were no statistically significant differences.

### 3.5. Reoperation

Among these patients who underwent mitral valve repair for endocarditis, seven (6.9%) ultimately required mitral valve reoperation at a median of 5 years after their initial operation. Of these, only two were for repeat IE, and the rest were for either repair failure, valve degeneration, and one was for obstructive coronary artery disease and underwent coronary artery bypass grafting. These patients were of a mean age of 35.9 ± 7.3 years (range 22–44 years) at the time of their initial operation. Overall, 5- and 10-year freedom from mitral valve reoperation was 93.6 ± 3.4% and 87.7 ± 5.2%, respectively. Freedom from reoperation is shown visually in Figure 1. 

## 4. Discussion

In this study, we evaluated the long-term outcomes of a large cohort of patients who underwent mitral valve repair for endocarditis at our institution. Early and late outcomes were favourable, including a perioperative mortality occurring in only one patient, a 5-year survival of 80.8 ± 4.7%, and a 10-year survival of 61.2 ± 9.2%. There was also a low incidence of mitral valve reoperation late after surgery. Of the seven patients who required mitral valve reoperation, reoperative surgery was performed at a median of 5 years after initial surgery. 

These data compare favourably with the current published literature. A retrospective observational study by Okada et al. included 171 consecutive patients who underwent mitral valve repair for endocarditis. They compared two groups, one with healed IE at the time of repair (n = 98) and one with active infection at the time of repair (n = 49). Freedom from reoperation at 5 years was 99% in the healed group and 89.6% in the active IE group; event-free survival at 10 years was 89.6% for the healed group and 72.6% for the active IE group [18]. Iung et al. reported short- and mid-term outcomes in a cohort of 63 patients who underwent mitral valve repair for native valve endocarditis at their center [19]. They report two deaths and two reoperations in the early postoperative period (within 30 days). Their 7-year event-free survival was 78 ± 6%. Patients in the acute endocarditis group experienced more postoperative events during follow-up compared with those operated on with healed IE but both groups had good late survival [19]. Kanemitsu et al. conducted a retrospective study reviewing 43 patients who underwent mitral valve repair for acute mitral valve endocarditis at their center with a mean follow-up of 7.4 ± 4.6 years [20]; there were no in-hospital deaths among this group of patients. The 5- and 10-year actuarial survival rates were 92 ± 4.1% and 83.5 ± 7.3%, respectively. The 5- and 10-year actuarial rates of freedom from reoperation were 90.5 ± 4.5% and 86 ± 5.8%, respectively. Overall, these outcomes are similar to our cohort of patients at our center. Our data include a larger cohort of patients and adds to the limited research on long-term results following mitral valve repair for mitral valve endocarditis. 

Other observational studies have shown good short- and mid-term event-free survival and freedom from reoperation outcomes following mitral valve repair IE [10,19,21,22,23]. It is important to note that, for the most part, outcomes are worse compared with pure degenerative mitral valve disease given the infective nature of the disease, along with its risk factors of the IE population [3,4,5,18].

Notwithstanding, the ability to repair an infected mitral valve is dependent on the extent of disease. The 2023 ESC guidelines for the management of endocarditis state that mitral valve repair should only be attempted if there can be durable repair with full elimination of infected tissue, and cannot be concluded as superior to replacement due to the high probability of selection bias [12]. The paucity of data on long-term outcomes also highlights the need for routine, systematic follow-up with echocardiography to better inform clinical practice and improve patient care [15]. Our study shows insights into the long-term outcomes following mitral valve repair at our institution; however, it does not compare the long-term outcomes with mitral valve replacement.

Awad et al. conducted the most up-to-date systematic review and meta-analysis, including 23 relevant publications for a total population of 11,802, comparing patients that underwent mitral valve repair versus mitral valve replacement for IE [8]. The meta-analysis favours mitral valve repair for long-term survival and lower risk of recurrence and freedom from reoperation. Repair was also superior to replacement in terms of in-hospital mortality. Similar results in a meta-analysis conducted by Harky et al. showed the mitral valve repair group had shorter cardiopulmonary bypass time, decreased bleeding, and decreased recurrence of endocarditis in the short-term (within 30 days) compared with the mitral valve replacement group. Additionally, there was a statistically significant improvement in survival at 1- and 5-year postoperative time-frames in the mitral valve repair group [24]. 

Toyoda and colleagues conducted a large, multi-center, retrospective cohort study comparing the outcomes of patients undergoing mitral valve repair versus replacement for native valve IE in a total of 164 hospitals in New York State and California State [14]. The propensity-matched cohort included 266 patients in the repair group and 532 in the replacement group, with a median follow-up of 7.2 years in the repair group and 6.5 years in the replacement group. They report an increased 30-day mortality in patients who underwent mitral valve replacement compared with repair, as well as significantly increased 90-day mortality. Mitral valve repair was also associated with better 12-year survival compared with replacement. Repair was also associated with lower rates of recurrent IE compared with the replacement group. There were no significant differences in rates of reoperation. 

In a recent retrospective single-center study by Di Bacco et al., they investigated the short- and long-term outcomes of 109 consecutive patients with acute mitral valve IE who underwent either mitral valve repair (n = 53) or replacement (n = 56). There were no statistically significant differences in in-hospital mortality and 10-year survival between the two groups; however, the repair group had lower incidences of IE relapse compared with the mitral valve replacement group, and IE relapse was determined an independent risk factor for mortality [7]. 

Furthermore, many observational studies show that there is decreased event-free survival in the mid- and long-term in the mitral valve repair group compared with replacement for mitral valve endocarditis [6,23,25,26,27]. Overall, the literature largely favours mitral valve repair for endocarditis whenever possible if there is enough valve tissue preserved after infectious tissues have been resected. While our paper does not focus on comparing the outcomes of repair versus replacement, it is important to recognize this as a future comparison where research can be explored to compare the long-term outcomes between repair and replacement, especially at an institution that has expert surgeons and case volume in mitral valve repair.

Lastly, our data showed a trend towards females being associated with higher risk of long-term mortality following mitral valve repair for mitral valve infective. Nonetheless, there were no statistically significant differences between females and males for long-term survival in our single-center cohort. While our cohort of patients did not show significant differences in survival between males and females, there are sex-based differences in survival noted in the literature for mitral valve repair for infective, as well as in mitral valve replacement. 

It has been well-noted in the literature that females fare worse in mitral valve disease and are underrepresented in many clinical trials and observational studies. This is also exemplified in our study, where only one-third of patients who underwent mitral valve repair at our institution were female. They often experience delayed diagnosis and increased postoperative complications as well as worse clinical outcomes following mitral valve surgery [28,29,30,31,32,33]. Severe mitral regurgitation is also more likely to be underestimated in females compared with males due to decreased ventricular dimensions and regurgitant volumes, which can lead to delayed surgical interventions [33,34,35]. Surgical mitral valve repair is also more difficult as females are less likely to have reparable leaflets [33]. In the general mitral valve repair population, not specific to patients with IE, females have inferior surgical outcomes such as worse long-term survival compared to their male counterparts [30,33]. 

In a systematic review by Slouha et al., they reported 34 publications that investigated the sex-based differences in IE between males and females [34]. Females were more likely to be younger with mitral valve involvement, and to have atrial fibrillation, chronic kidney disease, psychiatric disorders, and to be taking immunosuppressants. Additionally, they found that females were more likely to have culture-negative IE [35,36], which contributes to delays in diagnosis and management of the disease. Females were also associated with delayed presentation to seek medical help as well as more conservative treatment through antibiotic therapy as opposed to males, who were more likely to undergo surgical treatment. Females with IE were also associated with increased 30-day and 1-year mortality rates. 

There seems to be limited research on the sex-based differences in mitral valve surgery for IE, as well as few that report long-term outcomes comparing males and females. A study by Curlier and colleagues investigated the sex-based differences in patients who underwent early valve surgery for left-sided IE in a population-based cohort study [37]. They included 466 male and 154 female patients who underwent early valve surgery for left-sided IE. There were fewer women in this cohort compared with males. They found that even on initial presentation of endocarditis, women were significantly older as well as having increased preoperative comorbidities, making them more complex operative candidates. They were also less likely to undergo early valve surgery, but sex was not an independent predictor of early valve surgery. Moreover, while sex was not an independent predictor of 1-year mortality, women had a significantly higher risk of early postoperative mortality compared with men. However, it is difficult to conclude the implications of these findings in the grand scheme of surgical management of IE. They do not report the complications that occurred after early valve surgery and do not report on the reasons for delayed early valve surgery in the female cohort. There could have been an absence of indication for surgery or other factors that would have made surgery a contraindication. This study was also not specifically reporting specifically on mitral valve IE, but rather, all cases of endocarditis involving a left-sided heart valve. There are different surgical indications and implications with the aortic valve compared with mitral valve IE. Overall, while the results are consistent with what is reported in the literature of females having delayed surgery as well as worse postoperative outcomes, it is important to elucidate the reasons behind these delays as well as worse outcomes. 

In a cross-sectional study investigating 75 patients who underwent mitral valve repair for endocarditis at a single center, women were found to have significantly higher mortality, major adverse cardiac and cerebrovascular events, rate of reoperation, and treatment failure in comparison with men at 1-year follow-up [38]. Of note, women had significantly increased preoperative comorbidities of diabetes mellitus, hypertension, and hypercholesterolemia, whereas men were more likely to have a history of smoking. This highlights the difficulty in elucidating sex-based differences in mitral valve repair for native mitral valve IE when other comorbidities are at play preoperatively that could be the reasons for worse postoperative outcomes.

Leterrier et al. also report in-hospital mortality in females compared with males undergoing surgery for IE. They note that while females were associated with higher in-hospital mortality, multivariable analysis showed that female sex was not an independent determinant of in-hospital mortality. Rather, age, antibiotic use less than 7 days before surgery, and staphylococcal IE were independent determinants of in-hospital mortality [37]. 

Another study that used the National Inpatient Sample Database to identify patients who underwent surgical intervention for IE compared the differences in preoperative patient characteristics as well as postoperative clinical outcomes of patients who were hospitalized for IE stratified by sex [39]. A total of 81,942 hospitalizations were identified with a primary diagnosis of endocarditis, of which 36,302 (44%) were female. Women were older with decreased risk factors of drug abuse, congenital heart disease, hepatitis C, prosthetic valve endocarditis, prior valve replacement, prior percutaneous coronary intervention, prior coronary artery bypass grafting, and the presence of cardiac arrhythmias such as atrial fibrillation. Women were found to have increased comorbidities of controlled diabetes mellitus and controlled hypertension. Men also had higher incidence of myocardial infarction, cardiogenic shock, and need for mechanical ventilation compared with women. In this cohort, women were significantly less likely to undergo mitral valve replacement but there were no significant differences between men and women with regards to undergoing mitral valve repair. Among patients who underwent valve replacement surgery, women were associated with a statistically significant increase in in-hospital mortality after adjustment with multivariable logistic regression analysis. There is an increased use of valve surgery of IE in both men and women with improved trends in survival in both groups but there still exists sex-based disparities in survival between men and women. 

While our study does not show any statistically significant differences in mid- and long-term survival between males and females in our cohort of patients who underwent mitral valve repair for native mitral valve endocarditis, there was a trend correlating female sex with increased mortality. Only one-third of our study population were females, which is similar to other studies reporting mitral valve repair outcomes with women being underrepresented in the current research. It would be valuable, as we continue to enroll patients in our Surgical Mitral Valve Database, to further elucidate sex-based differences in preoperative, intraoperative, and postoperative outcomes following mitral valve repair on a short-, mid-, and long-term basis. Trends in sex-based differences are outlined but not fully understood, and further research can be performed to determine how to decrease the risks associated. 

## 5. Limitations

This is a single-center non-randomized experience; therefore, outcomes may not be generalizable to certain regions/catchments. Additionally, all surgeries were performed by two surgeons at a single center with extensive expertise and training in mitral valve repair. This decreases the external validity of these results. The retrospective study design is also a limitation of this study, as there were limitations on the data available to be analyzed of the patients included in our cohort. For example, while all patients underwent preoperative transesophageal echocardiography, vegetation size and extent of leaflet involvement were not available for analysis. Hence, this limits our analysis of the preoperative characteristics of our patients.

## 6. Conclusions

In conclusion, our study shows that mitral valve repair can be an effective method for treating IE, with low reoperation rates and mortality in the long term. It is important to assess the feasibility of repair pending the extent of infection, as well as to recognize the complexity of cases and for the mitral valve repair to be carried out in expert centers to provide robust and durable long-term outcomes. There can also be further research on the sex-based differences in outcomes following mitral valve repair and how to optimally address these disparities in outcomes. 

## Figures and Tables

**Figure 1 microorganisms-12-01809-f001:**
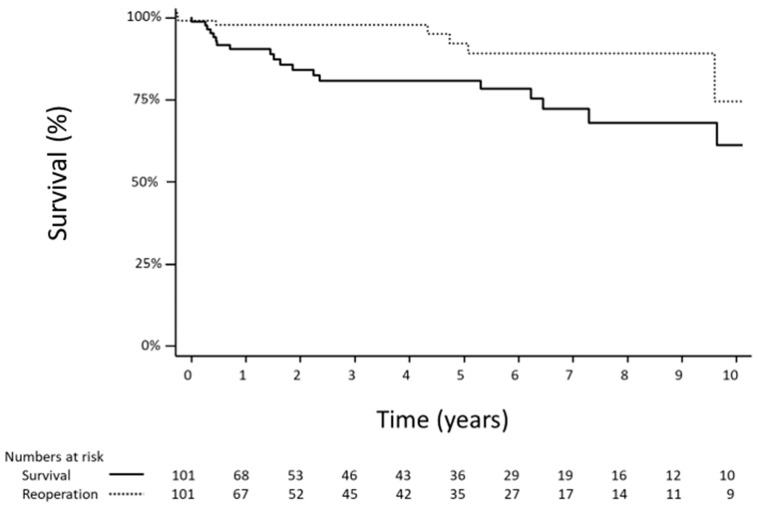
Total survival and freedom from reoperation in patients who underwent mitral valve repair for native valve infective endocarditis.

**Table 1 microorganisms-12-01809-t001:** Preoperative patient characteristics.

Variables	Total (N = 101)
Age (years)	55.3 ± 14.2
Sex (% female)	33 (33%)
Atrial fibrillation	13 (13%)
NYHA III/IV	13 (13%)
STS risk of mortality (%)	1.6 ± 2.1
LVEF (%)	61.5 ± 10.6
Mitral perforation	8 (7.9%)
Mitral regurgitation grade III/IV	83 (82%)
Concomitant coronary artery disease requiring CABG	9 (8.9%)

NYHA = New York Heart Association; STS = Society of Thoracic Surgeons; LVEF = left ventricular ejection fraction; CABG = coronary artery bypass grafting.

**Table 2 microorganisms-12-01809-t002:** Microbiological data from blood cultures.

Microorganism	Total (N = 101)
*Streptococcus viridans*	21 (21%)
Methicillin-susceptible *Staphylococcus aureus* (MSSA)	2 (2.0%)
*Staphylococcus aureus* (susceptibility unspecified)	5 (5.0%)
*Streptococcus gallolyticus*	1 (1.0%)
*Streptococcus* (species unspecified)	2 (2.0%)
*Enterococcus* (species unspecified)	2 (2.0%)
*Enterococcus durans*	1 (1.0%)
MSSA + *Enterococcus faecalis*	1 (1.0%)
*Haemophilus parainfluenzae*	1 (1.0%)
*Cardiobacterium hominis*	1 (1.0%)
*Escherichia coli*	1 (1.0%)
*Histoplasma capsulatum*	1 (1.0%)
*Corynebacterium*	1 (1.0%)
Coagulase-negative *Staphylococci*	2 (2.0%)
Negative blood cultures	22 (22%)
Not reported	37 (37%)

**Table 3 microorganisms-12-01809-t003:** Main indication for surgery.

Surgical Indication	Total (N = 101)
Heart failure with mitral regurgitation	89 (88%)
Persistent infection	2 (2.0%)
Locally uncontrolled	6 (5.9%)
Septic emboli	4 (4.0%)

**Table 4 microorganisms-12-01809-t004:** Surgical techniques used for mitral valve repair.

Surgical Technique	N
Neochords	25
Posterior leaflet resection	25
Anterior leaflet resection	13
Chordal transfer	21
**Annuloplasty**	
Futureband	95
Duran	1
Physio	4
Cosgrove	1
Annuloplasty size (mm)	30 ± 3
	Median: 30 (IQR 28–32)

IQR = interquartile range.

## Data Availability

Dataset available on request from the authors.

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
