# Peer review of "Longitudinal Outcomes Following Mitral Valve Repair for Infective Endocarditis"

_microorganisms, 2024, doi:10.3390/microorganisms12091809_

Round 1
Reviewer 1 Report
Comments and Suggestions for Authors
Well documented work
Nice argue for repair instead of change.
Author Response
Comment 1:
Well documented work
Nice argue for repair instead of change.
Response 1:
Thank you so much for this feedback for our manuscript!
Reviewer 2 Report
Comments and Suggestions for Authors
In the study by Yuan Qiu et al., conducted a retrospective, single-center study. It included patients who underwent mitral valve repair surgery for infective endocarditis between 2001 and 2021 at the University of Ottawa Heart Institute. It would be beneficial to compare these results with those from other reported studies.
Overall, the study suggests that mitral valve repair can be an effective method for treating infective endocarditis, offering favorable long-term outcomes in terms of freedom from reoperation and mortality.
Suggestions for Improvement:
1. Figure 1: The Y-axis needs to be labeled with "Survival" to make it clearer and easier to understand.
2. Table 2: Use "Previously healed infective endocarditis (IE)," delete the abbreviation.
3. Table 3: Clarify what "IQR" stands for. Providing a definition will help readers unfamiliar with the term.
4. Table 4 Covariate Analysis: It would be helpful to include an analysis of any significant covariates. This additional analysis could offer more insights into the factors influencing the outcomes.
Author Response
Response to Reviewer #2 Comments:
In the study by Yuan Qiu et al., conducted a retrospective, single-center study. It included patients who underwent mitral valve repair surgery for infective endocarditis between 2001 and 2021 at the University of Ottawa Heart Institute. It would be beneficial to compare these results with those from other reported studies.
Thank you so much for this feedback. We have included how our results compare to other reported studies in the discussion section.
Overall, the study suggests that mitral valve repair can be an effective method for treating infective endocarditis, offering favorable long-term outcomes in terms of freedom from reoperation and mortality.
Thank you for this feedback.
Suggestions for Improvement:
- Figure 1: the Y-axis needs to be labelled with “Survival” to make it clearer and easier to understand
Thank you for this suggestion. We have labelled the Y-axis with “Survival” (line 112).
- Table 2: Use "Previously healed infective endocarditis (IE)," delete the abbreviation.
Thank you for this feedback, we have actually removed this row in the table as now it is grouped as “Negative blood cultures” in Table 2 in response to another reviewer comment.
- Table 3: Clarify what "IQR" stands for. Providing a definition will help readers unfamiliar with the term.
Thank you for this suggestion, we have included “IQR = interquartile range” below Table 3.
- Table 4 Covariate Analysis: It would be helpful to include an analysis of any significant covariates. This additional analysis could offer more insights into the factors influencing the outcomes.
We thank the Reviewer for this important comment. We had included age, female gender, and preoperative atrial fibrillation as these have been previously described as risk factors associated with the outcome of interest. The low number of events in this study also limits the number of covariates that can be reasonably assessed. The risk factors listed above were included in a single adjusted Cox proportional hazards model. Below are the univariate point estimates. These have not be included in the manuscript as they do not change the study findings, but we can do so as required by the journal.
Table A. Risk factors associated with mortality
|
Covariate |
Hazard ratio ± standard error (p-value) |
|
Age |
1.00 ± 0.14 (p = 0.7) |
|
Female |
1.66 ± 0.76 (p = 0.3) |
|
Atrial fibrillation |
1.96 ± 1.11 (p = 0.2) |
*Variables were assessed via univariate analysis
Reviewer 3 Report
Comments and Suggestions for Authors
Longitudinal Outcomes Following Mitral Valve Repair for Infective Endocarditis PEER-REVIEW This is a single-centre observational retrospective study addressing the outcome of patients who underwent mitral valve repair for infective endocarditis. The topic is of interest. The title is clear and describe the focus of study. Overall, the manuscript is well written, and I did not detect major issues with grammar and English medical language. The introduction is clear. In my opinion, the are several important points that requires major revisions and these are listed below together with few minor changes suggested. My final recommendation is “reconsider of major revision” Major revisions: 1. Methods, IE definition: authors should state which criteria have been used to diagnose infective endocarditis during the inclusion time interval and add linked references. Moreover, they should state if only definite IE have been included. 2. Methods, statistics: authors should better describe the Cox regression model. Which variables were included in the model? Was it preceded by a univariate analysis? If yes, authors included in the multivariable Cox regression model only those variables significantly associated with mortality at univariate or also those not significantly associated but clinically relevant? Was it adjusted or non-adjusted? These points should be clarified. 3. Results, cardiac surgery indication: authors should categorize the indication for cardiac surgery according to most recent guidelines (heart failure vs uncontrolled infection vs high risk of embolism). 4. Results, Table 1: concomitant CABG should be mentioned in the “cardiac surgery procedures” section and not in the “preoperative characteristics”. 5. Results, preoperative echocardiographic data: if available, authors should more carefully report echocardiographic data. How many patients underwent transesophageal echocardiography? Vegetations number, maximum length and leaflet location?. If these data are not available it should be mentioned among the limitations. 6. Results, infective endocarditis microorganisms: several changes should be applied to this section. Title: Etiology or causative microorganisms or microbiological isolates. I suggest do not use the term bacteraemia in this section because it refers to uncomplicated positive blood cultures. Authors could use the terms: Streptococcal IE or “The causative microorganism was Streptococcus spp. in XX cases”. Table 2 – Streptococcus mitis and anginosus falls into the Streptococcus viridans group. Change results accordingly. Change Streptococcus bovis with Streptococcus gallolyticus (current terminology). Staphylococcus lugdunensis falls into the coagulase negative Staphylococci. If authors want to report it separately, then they should change the name of the group “coagulase-negative Staphylococci” in “other coagulasenegative Staphylococci”. Libman-Sacks and Previously healed cannot be listed among “microorganisms” – please consider to list them in the group “Negative blood cultures”. Histoplasmosis refers to the disease – If Histoplasma capsulatum was isolated from blood cultures then the name of the pathogen should be reported. 7. Results, multivariable Cox regression model: as previously said in point 2 the authors should provide results of univariate analysis preceding the Cox regression highlighting all the variables included in the model. Table 4 should be modified including also those variables included in the model but not significantly associated with the outcome. 8. Discussion section: the discussion section is too long and is out of the scope of the study even though it is providing a comprehensive narrative review. Moreover, authors do focus on description of current studies without providing comparison with their results. Therefore, authors could consider changing the article methodology adding a narrative literature review (in this case I suggest providing a table that summarize the main findings of the most important studies) or shorten the discussion section by including only those studies in which a direct comparison with their results is possible and clinically relevant. 9. Limitation section: this section should be extensively revised focusing on limitations of the study. Lines 358-361 should be deleted. 10. Abstract: As stated by the authors in the introduction section line 36-39 there are insufficient evidence regarding comparison between mitral valve repair and replacement and, at the moment, repair cannot be considered superior to replacement and vice versa. Therefore the first sentence of the abstract “Mitral valve repair is the ideal approach in managing mitral valve infective endocarditis for patients requiring surgery” should be rephrased accordingly. 11. Ethics: data regarding ethics approval should be clarified (ethics committee approval? Number?. Information regarding informed consent is not shown. Minor changes: 1. Line 60: “categorical variable as described” – please amend it 2. Line 230: “eh”, please amend it. 3. Consider using abbreviations such as “infective endocarditis – IE”

Author Response
Response to Reviewer #3 Comments:
This is a single-centre observational retrospective study addressing the outcome of patients who underwent mitral valve repair for infective endocarditis.
The topic is of interest. The title is clear and describe the focus of study. Overall, the manuscript is well written, and I did not detect major issues with grammar and English medical language. The introduction is clear.
In my opinion, the are several important points that requires major revisions and these are listed below together with few minor changes suggested.
My final recommendation is “reconsider of major revision”
Thank you so much for your feedback. We have addressed each recommendation below.
Major revisions:
- Methods, IE definition: authors should state which criteria have been used to diagnose infective endocarditis during the inclusion time interval and add linked references. Moreover, they should state if only definite IE have been included.
These patients were diagnosed with infective endocarditis based on Duke criteria considering assessment by the treatment team. The indication for surgery were heart failure symptoms with mitral regurgitation in 87 patients, persistent infection in 2 patients, prosthetic valve endocarditis in 2 patients, locally uncontrolled IE in 6 patients, and septic emboli in 4 patients.
- Methods, statistics: authors should better describe the Cox regression model. Which variables were included in the model? Was it preceded by a univariate analysis? If yes, authors included in the multivariable Cox regression model only those variables significantly associated with mortality at univariate or also those not significantly associated but clinically relevant? Was it adjusted or non-adjusted? These points should be clarified.
We thank the Reviewer for this important comment. We had included age, female gender, and preoperative atrial fibrillation as these have been previously described as risk factors associated with the outcome of interest. The low number of events in this study also limits the number of covariates that can be reasonably assessed. The risk factors listed above were included in a single adjusted Cox proportional hazards model. Below are the univariate point estimates. These have not be included in the manuscript as they do not change the study findings, but we can do so as required by the journal.
Table A. Risk factors associated with mortality
|
Covariate |
Hazard ratio ± standard error (p-value) |
|
Age |
1.00 ± 0.14 (p = 0.7) |
|
Female |
1.66 ± 0.76 (p = 0.3) |
|
Atrial fibrillation |
1.96 ± 1.11 (p = 0.2) |
*Variables were assessed via univariate analysis
- Results, cardiac surgery indication: authors should categorize the indication for cardiac surgery according to most recent guidelines (heart failure vs uncontrolled infection vs high risk of embolism).
Thank you for this important comment. We have added a paragraph and Table 3 in section 3.3 relabeled “Surgical Indication and Technique” that summarizes the indication for cardiac surgery according to the 2023 European Society of Cardiology Guidelines for the management of endocarditis.
- Results, Table 1: concomitant CABG should be mentioned in the “cardiac surgery procedures” section and not in the “preoperative characteristics”.
Thank you for this comment. We have revised this to “concomitant coronary artery disease requiring CABG” as the “surgical techniques” table is more focused on the mitral valve repair surgical technique, and the concomitant CABG was to communicate how many patients required concomitant CABG at the time of their mitral valve surgery.
- Results, preoperative echocardiographic data: if available, authors should more carefully report echocardiographic data. How many patients underwent transesophageal echocardiography? Vegetations number, maximum length and leaflet location?. If these data are not available it should be mentioned among the limitations.
All patients underwent preoperative transesophageal echocardiography. The indications for surgery included heart failure, persistent infection, prosthetic valve IE, locally uncontrolled IE, septic emboli, and severe mitral regurgitation. Unfortunately, vegetation size and extent of leaflet involvement is not available for analysis.
- Results, infective endocarditis microorganisms: several changes should be applied to this section. Title: Etiology or causative microorganisms or microbiological isolates. I suggest do not use the term bacteraemia in this section because it refers to uncomplicated positive blood cultures. Authors could use the terms: Streptococcal IE or “The causative microorganism was Streptococcus spp. in XX cases”.
Thank you for this feedback. The title has been changed to “Causative Microorganisms”. We changed the subsequent section in 3.2 to reflect this language.
Table 2 – Streptococcus mitis and anginosus falls into the Streptococcus viridans group. Change results accordingly. Change Streptococcus bovis with Streptococcus gallolyticus (current terminology). Staphylococcus lugdunensis falls into the coagulase negative Staphylococci. If authors want to report it separately, then they should change the name of the group “coagulase-negative Staphylococci” in “other coagulase-negative Staphylococci”. Libman-Sacks and Previously healed cannot be listed among “microorganisms” – please consider to list them in the group “Negative blood cultures”. Histoplasmosis refers to the disease – If Histoplasma capsulatum was isolated from blood cultures then the name of the pathogen should be reported.
Thank you so much for these comments. The appropriate changes have been made according to this feedback in Table 2. Histoplasma capsulatum was isolated from cultures and the row name has been changed to reflect that.
- Results, multivariable Cox regression model: as previously said in point 2 the authors should provide results of univariate analysis preceding the Cox regression highlighting all the variables included in the model. Table 4 should be modified including also those variables included in the model but not significantly associated with the outcome.
Thank you for this important comment. We had included age, female gender, and preoperative atrial fibrillation as these have been previously described as risk factors associated with the outcome of interest. The low number of events in this study also limits the number of covariates that can be reasonably assessed. The risk factors listed above were included in a single adjusted Cox proportional hazards model. Below are the univariate point estimates. These have not be included in the manuscript as they do not change the study findings.
Table A. Risk factors associated with mortality
|
Covariate |
Hazard ratio ± standard error (p-value) |
|
Age |
1.00 ± 0.14 (p = 0.7) |
|
Female |
1.66 ± 0.76 (p = 0.3) |
|
Atrial fibrillation |
1.96 ± 1.11 (p = 0.2) |
*Variables were assessed via univariate analysis
- Discussion section: the discussion section is too long and is out of the scope of the study even though it is providing a comprehensive narrative review. Moreover, authors do focus on description of current studies without providing comparison with their results. Therefore, authors could consider changing the article methodology adding a narrative literature review (in this case I suggest providing a table that summarize the main findings of the most important studies) or shorten the discussion section by including only those studies in which a direct comparison with their results is possible and clinically relevant.
Thank you for this feedback. We have shortened the discussion section by including only those studies in which a direct comparison with our results is possible and clinically relevant, thank you.
- Limitation section: this section should be extensively revised focusing on limitations of the study. Lines 358-361 should be deleted.
Thank you for this feedback. The indicated lines are deleted and this section has been revised to focus on limitations of the study.
- Abstract: As stated by the authors in the introduction section line 36-39 there are insufficient evidence regarding comparison between mitral valve repair and replacement and, at the moment, repair cannot be considered superior to replacement and vice versa. Therefore the first sentence of the abstract “Mitral valve repair is the ideal approach in managing mitral valve infective endocarditis for patients requiring surgery” should be rephrased accordingly.
Thank you for this comment. We have changed to “mitral valve repair is the preferred approach in managing mitral valve infective endocarditis”.
- Ethics: data regarding ethics approval should be clarified (ethics committee approval? Number?. Information regarding informed consent is not shown.
Thank you for this comment, we have included the ethics committee approval and approval number, as well as a statement about informed consent in section 2.1: “Ethics approval was obtained through the University of Ottawa Heart Institute Research Ethics Board (approval number 20160395-01H) and is active for longitudinal assessment of outcomes following mitral valve repair. Informed consent was obtained for all patients undergoing surgery.”
Minor changes:
- Line 60: “categorical variable as described” – please amend it
Thank you, this has been amended.
- Line 230: “eh”, please amend it.
Thank you, this has been amended to “the”.
- Consider using abbreviations such as “infective endocarditis – IE”
Thank you for this feedback, we have abbreviated “infective endocarditis” to “IE” throughout the manuscript.
Round 2
Reviewer 3 Report
Comments and Suggestions for Authors
Please see attached file

Author Response
Microorganisms Response to Reviewers’ Comments
Title: Longitudinal Outcomes Following Mitral Valve Repair for Infective Endocarditis
We thank the editorial team for their time and effort in reviewing our manuscript. We have provided detailed responses to the reviewers’ comments below, with revisions in the manuscript using the “Track Changes” feature in Microsoft Word.
Response to Reviewer #3 Comments:
The authors made a great job in assessing most of my major concerns. However, I believe that point 1, 2 and 8 of the major revision section require further attention (as detailed below) and I recommend to “reconsider after major revision”.
Thank you so much for your feedback. We have addressed each recommendation below.
1 – R2: To ensure data reproducibility the criteria used to define IE in this cohort need to be stated in the methods section together with the associated references. PVE by itself is not an indication for cardiac surgery if not caused by S. aureus or non-HACEK gram negative bacilli (ESC 2023). Please specify.
We thank the Reviewer for this important comment. The definition has now been included with the associated reference at the beginning of the “3.3 Surgical Indication and Technique” section. Additionally, upon further review of the 2 patients with PVE their main indication for surgical intervention was HF with MR, so that has been updated accordingly as well in section 3.3.
2 – R2: Female gender and preoperative atrial fibrillation are not clear independent predictors of poor outcome among IE patients (as also shown in the ESC 2023 guidlelines – supplementary data Table S7) with previous studies providing conflicting results. The reason why these variables have been included in a multivariate model is unclear and conclusion regarding these results are not straightforward. Authors may consider deleting this analysis from the manuscript. If authors decide to include this in the final manuscript, I believe that the methodology supporting it has to be clearly reported in the methods section (statistical analysis).
We thank the Reviewer for this important remark. Although female gender and atrial fibrillation have not always been associated with mortality in patients with infective endocarditis, we have shown that these risk factors have been predictive of survival in patients with degenerative disease. Notwithstanding the Reviewer suggestion, we have removed the univariate and multivariate tables and included these separately as supplemental tables.
8 – R2: The discussion section has been shortened of approximately 20 lines and it does not change the overall length of this section. Also, the text has not changed, and authors are still focusing on a detailed description of previous studies results rather than a comparison of their results with current literature. For instance, Line 180 – 230 is focusing on the description of studies comparing mitral valve repair vs mitral valve replacement. This is not addressed by the authors in this study. I believe that this point requires further attention.
Thank you for this important feedback. We have added more focus on relating and comparing our results with the current literature throughout the discussion. We have now removed most of the text (lines 194 – 215) that discusses the literature on mitral valve repair versus replacement and included a sentence that does discuss “While our paper does not focus on comparing the outcomes of repair versus replacement, it is important to recognize this as a future comparison where research can be explored to compare the long-term outcomes between repair and replacement, especially at an institution that has expert surgeons and case volume in mitral valve repair.” However, if you feel that we should remove the mitral valve repair versus replacement text altogether also happy to do that.
Respectfully submitted,
Yuan and Vince